# Development of Time-Weighted Average Sampling of Odorous Volatile Organic Compounds in Air with Solid-Phase Microextraction Fiber Housed inside a GC Glass Liner: Proof of Concept

**DOI:** 10.3390/molecules24030406

**Published:** 2019-01-23

**Authors:** Madina Tursumbayeva, Jacek A. Koziel, Devin L. Maurer, Bulat Kenessov, Somchai Rice

**Affiliations:** 1Department of Agricultural and Biosystems Engineering, Iowa State University, Ames, IA 50011, USA; madina@iastate.edu (M.T.); dmaurer@iastate.edu (D.L.M.); somchai@iastate.edu (S.R.); 2Department of Meteorology and Hydrology, Al-Farabi Kazakh National University, Almaty 050040, Kazakhstan; 3Center of Physical Chemical Methods of Research and Analysis, Al-Farabi Kazakh National University, Almaty 050012, Kazakhstan; bkenesov@cfhma.kz

**Keywords:** SPME, retracted SPME, TWA SPME, GC–MS, on-site sampling, air quality, air monitoring, VOCs, odor, environmental analysis

## Abstract

Finding farm-proven, robust sampling technologies for measurement of odorous volatile organic compounds (VOCs) and evaluating the mitigation of nuisance emissions continues to be a challenge. The objective of this research was to develop a new method for quantification of odorous VOCs in air using time-weighted average (TWA) sampling. The main goal was to transform a fragile lab-based technology (i.e., solid-phase microextraction, SPME) into a rugged sampler that can be deployed for longer periods in remote locations. The developed method addresses the need to improve conventional TWA SPME that suffers from the influence of the metallic SPME needle on the sampling process. We eliminated exposure to metallic parts and replaced them with a glass tube to facilitate diffusion from odorous air onto an exposed SPME fiber. A standard gas chromatography (GC) liner recommended for SPME injections was adopted for this purpose. Acetic acid, a common odorous VOC, was selected as a model compound to prove the concept. GC with mass spectrometry (GC–MS) was used for air analysis. An SPME fiber exposed inside a glass liner followed the Fick’s law of diffusion model. There was a linear relationship between extraction time and mass extracted up to 12 h (*R*^2^ > 0.99) and the inverse of retraction depth (1/*Z*) (*R*^2^ > 0.99). The amount of VOC adsorbed via the TWA SPME using a GC glass liner to protect the SPME was reproducible. The limit of detection (LOD, signal-to-noise ratio (S/N) = 3) and limit of quantification (LOQ, S/N = 5) were 10 and 18 µg·m^−3^ (4.3 and 7.2 ppbV), respectively. There was no apparent difference relative to glass liner conditioning, offering a practical simplification for use in the field. The new method related well to field conditions when comparing it to the conventional method based on sorbent tubes. This research shows that an SPME fiber exposed inside a glass liner can be a promising, practical, simple approach for field applications to quantify odorous VOCs.

## 1. Introduction

Offensive odors dispersed from animal feeding operations are a common concern for neighboring communities [1]. These odors originate mainly from manure and other organic matters in livestock operations and are a complex mixture of many gases, of which the largest portion (by number) are volatile organic compounds (VOCs). VOCs are complex chemicals distinguished by their ability to evaporate easily at room temperatures. VOCs originating from industry and transportation are studied extensively. Less attention is focused on VOCs found in animal production systems. However, the research in this area, especially research on the mitigation of odor emissions, is still limited [1,2,3,4,5] especially in regards to farm-scale proven technologies. Public concerns and research interest are focused mainly on solving odor nuisance.

Addressing public concerns about odorous emissions from livestock operations is challenging since many of these VOCs usually have a low odor detection threshold. Even at low concentrations (ppbV, pptV), they can be potent and objectionable odorants [6]. Thus, sampling and analysis of VOCs associated with animal operations are still challenging. Methods to detect and quantify VOCs from animal facilities are important for measuring air quality, developing and testing technologies that can mitigate odorous emissions. Many approaches used for sampling and analysis of VOCs are effective for qualitative analysis, but many standard methods developed for urban air are typically either not suitable for typical odorous VOCs or not sensitive enough to quantify trace concentrations.

Numerous VOCs can be found at animal facilities. Starting from 1965 when stearic acid was first identified [7], the list of known VOCs at animal facilities is constantly expanding. The results of the most recent studies show that more than 512 VOCs in total are found at swine facilities [7]. VOCs found in animal facilities can be classified into several groups. They are acids, alcohols, aldehydes, amines, hydrocarbons, indoles, nitrogen-containing compounds, phenols, sulfur-containing compounds, volatile fatty acids, and others [8]. However, sulfur-containing VOCs (S-VOCs) and volatile fatty acids (VFAs) were identified as the most dominant classes of VOCs at animal facilities which are responsible for those offensive odors [6]. A derivative of phenolics, *p*-cresol, was reported to be one of the main compounds responsible for characteristic odor at swine barns [6,9]. In order to test sampling methods, most studies focused on 10–15 odorous VOCs, which were used to sample emissions from livestock farms or to simulate them in a laboratory [6,10,11]. Some of the odorous VOCs include acetic, propionic, butyric, and isovaleric acids; methyl, ethyl, and butyl mercaptans; dimethyl sulfide, *p*-cresol, and others.

Acetic acid is considered the most abundant VOC in any animal facility, including swine farms. It is a colorless liquid that can be easily evaporated, and it has a strong and distinct pungent and vinegar-like smell. It was reported that the concentration of acetic acid in gaseous emissions from swine and dairy farms in the United States (US) could range from ~1 to 617 mg·m^−3^ [11]. Due to its abundance, it is reasonable to consider acetic acid as a model compound to validate concepts involving new VOC sampling methods and for testing the effectiveness of odorous VOC mitigation technologies in the context of livestock agriculture.

### 1.1. Air Sampling of Odorous VOCs

Most odorous VOCs are found at low concentrations [11]. VOC quantification requires reliable air sampling techniques and analytical methods that are representative of the air at the monitored site. The time-weighted average (TWA) sampling approach can be useful in such cases. This approach is used to determine the average concentration of an air pollutant over periods that can extend from a few minutes to several weeks [12]. TWA concentrations are needed to estimate average exposure to a contaminant. A number of different sampling techniques were introduced to obtain TWA concentrations of VOCs in the field. To date, the most common techniques are whole-air sampling techniques and sorbent tubes [13,14]. A short summary of those methods is given in Appendix A. Those methods require specialized equipment [14,15,16,17,18,19,20,21] (cleaning and evacuation of canisters, flushing air sampling bags with ultra-pure air or nitrogen, thermal desorption, air sampling pump) which makes the methods laborious and expensive to work with. Thus, simpler and more reliable methods to quantify VOCs at animal feeding operations are needed.

### 1.2. The TWA SPME Approach

Solid-phase microextraction (SPME) combines passive air sampling and sampling preparation. SPME uses a compact-size sampler that consists of a polymeric fiber that is kept inside a hollow metallic needle. During air sampling, VOCs are collected on an SPME fiber. SPME was shown to provide low detection limits reaching parts-per-trillion levels. After VOCs are transferred to an analytical instrument (e.g., via hot gas chromatography (GC) injector), extracted VOCs are thermally desorbed from the fiber, which can be reused. Thus, SPME eliminates the need for solvents and works with existing analytical technologies.

SPME is applicable for assessment of TWA concentrations in continuous sampling mode where the SPME fiber is retracted into the needle at a known distance during the desired sampling time. In contrast to the exposed fiber where an analyte reaches an equilibrium with the SPME, extraction of VOCs via the retracted fiber is controlled by diffusion. Since the fiber is kept inside the needle and extraction of VOCs is controlled by diffusion, the extraction rates are lower. Thus, the fiber in the protecting needle can be used for longer periods before reaching an equilibrium with the environment [22]. Analytes accumulated on the SPME fiber enable the measurement of the average gas (e.g., a VOC or total VOCs) concentration to which the fiber was exposed [23].

Quantification of the TWA concentrations with a retracted SPME fiber follows Fick’s first law of diffusion (Equation (1)): the mass extracted on the fiber is proportional to (1) the diffusion coefficient of the analyte (*D_g_*), (2) the concentration of the analyte in the gas phase (*C_gas_*), (3) sampling time (*t*), and (4) cross-sectional area of the SPME needle opening (*A*); it is inversely proportional to the diffusion path length (*Z*, i.e., the distance from the needle opening to the tip of retracted fiber).
(1)$$n=D_{g}\frac{A}{Z}\int C_{gas}\left(t\right)dt$$

### 1.3. Application of the TWA SPME for VOCs

Despite the advantages of the TWA SPME approach, comparatively few studies were conducted to bring the approach to the field. The studies [12,24,25,26,27,28,29,30] showed that SPME devices could be used as TWA samplers to access exposure to different volatile (hydrocarbons, formaldehyde, and others) and chlorinated semi-volatile organic compounds [12] at the source. VOCs were also quantified from biomass gasification process streams in fast-moving environments at elevated temperatures such as syngas stream [28,29] and idling vehicle exhaust [30,31]. A major challenge with the TWA SPME approach is the influence of the metallic SPME needle assembly on the VOC extraction process, as documented earlier [28,29,30,31]. The metallic surface of the SPME needle (studied using “broken fiber”, i.e., fiber without coating) had adsorptive properties that were significant compared with the adsorption by the fiber itself. Similarly, Koziel et al. [32] evaluated the contribution of the metallic parts first before quantifying five biomarkers (VOCs) such as dimethyl disulfide, dimethyl trisulfide, pyrimidine, phenol, and *p*-cresol emitted during aerobic digestion of animal tissue. The current suggestion to overcome this issue for the TWA SPME for quantification of VOCs is the mandatory evaluation of the contribution of mass extracted by a “broken fiber” so this effect can be accounted for. Thus, while reproducible, the contribution of metallic SPME parts on the TWA SPME process adds more steps to method development.

Recent modeling of the TWA SPME process by Kenessov et al. [22] provided an insightful identification of limitations for the use of retracted SPME fibers and possible means to address them. In their study, they found that a Carboxen/polydimethylsiloxane (Car/PDMS) SPME fiber with a greater size of a protecting needle (23 ga; as opposed to 24 ga) extracted greater amounts of analytes (about 19% more) than the fiber with a smaller protecting needle gauge size. This study suggests that the space between the SPME coating and the inner wall of the protecting needle plays a crucial role in extracting mass, since it allows faster diffusion of analytes not only to the tip of the fiber but also to its sides. The paper also recommends using a 23-ga SPME fiber for quantification of analytes with lower detection limits. However, no research reported the quantification of major VOCs that are responsible for the characteristic offensive odor downwind from animal feeding operations using a TWA SPME approach.

In this research, we aimed at addressing two major needs and gaps in knowledge: (1) to minimize or eliminate the need to consider the effect of the metallic SPME needle on air sampling of VOCs with TWA SPME, and (2) to enable SPME technology to be used for odorous VOC quantification in farm environments.

### 1.4. Objectives

The goal of this work was to develop a method for the quantification of target odorous VOC (using acetic acid as a model compound) with a TWA SPME approach that is more accurate and less laborious. Unlike the previous TWA SPME approaches where an SPME fiber is retracted into a metallic needle, this research proposes to use an SPME fiber that is exposed inside the GC glass liner to achieve the effect of a traditional retracted fiber *without the need to estimate and account for the inherent adsorption of VOCs onto metallic parts of SPME needle during sampling*. Since a GC glass liner has a greater cross-sectional area than a traditional retracted SPME fiber (Figure 1), the new approach should allow for greater amounts of the analyte extracted on the fiber, and increased exposure of the side surfaces of the coating to the sample, resulting in lower detection limits and greater accuracy.

Our working hypothesis is that the glass liner enclosure might be less affected by the apparent departure from the ideal quantification model (Fick’s Law, Equation (1)) that is associated with the use of metallic needle enclosures to facilitate TWA SPME. The inside of the glass liner serves as a diffusion path. Thus, extraction of VOCs is controlled by diffusion, and potentially can be used for sampling of VOCs in remote locations. The method utilizes GC glass liners that are readily available in many analytical laboratories. As the most abundant VOC in livestock operations, acetic acid was chosen as a model compound to prove the concept.

The specific objectives of this research were to (1) build and verify a standard gas generation system for odorous VOC that simulates typical dynamic animal facility air in the lab; (2) test the performance of an SPME fiber retracted into a glass liner and the adherence of this air sampling concept to the Fick’s Law; (3) test the new method for quantification of acetic acid on a typical Iowa swine facility and evaluate its feasibility; and (4) compare the developed method side-by-side to a standard method under field conditions.

## 2. Results and Discussions

### 2.1. Standard Gas Stability Check

The stability of standard gas generated by the standard gas generation system is shown in Figure 2. For the purpose of checking stability, the standard gas was simultaneously measured with exposed and “retracted” SPME fibers and sorbent tubes several times per day for three consecutive days.

The result of daily extractions with exposed and “retracted” SPME fibers and sorbent tubes shows that the standard gas generation system was successful in generating a continuous supply of acetic acid. As can be seen in Figure 2, the exposed SPME fiber responses were more variable (relative standard deviation, RSD 5.6%) than the “retracted” SPME fiber (RSD 3.2%) in terms of extracted mass. Because the exposed SPME fiber was fully in contact with the moving gas, it resulted in more than two orders of magnitude higher extraction rates than the “retracted” SPME fiber. These results are consistent with the findings from Baimatova et al. [30]. The limits of detection (LOD, signal-to-noise ratio (S/N) = 3) and limits of quantification (LOQ, S/N = 5) were 10 and 18 µg·m^−3^ (4.3 and 7.2 ppbV), respectively.

### 2.2. Effects of SPME Fiber Type on TWA Sampling with Glass Liner

A glass liner facilitating TWA SPME was used. Two adsorptive SPME coatings were tested, i.e., 85-µm Car/PDMS and 50/30-µm (divinylbenzene, DVB) DVB/Car/PDMS, and both types of coatings effectively extracted acetic acid (Figure 3) for up to 12 h. Mass extracted by the fibers showed a linear response with sampling time (*R*^2^ > 0.99). However, the results show that the average masses extracted by both SPME fibers were higher than the theoretical value (Equation (1)) by 11.1% and 3.7% on average for Car/PDMS and DVB/Car/PDMS fibers, respectively. This (relatively small and reproducible) discrepancy from theory (Equation (1)) could be considered excellent, considering that no effects of metallic SPME fiber assembly were taken into account.

The 85-µm Car/PDMS fiber provided a slightly higher response than the DVB/Car/PDMS fiber for acetic acid, which is consistent with the studies of Kenessov et al. [22] and Abalos et al. [33]. The total mass extracted by the SPME fibers was reproducible. The RSDs of MS responses with Car/PDMS (ranging from 2.3% to 12.2%) were lower in comparison with the DVB/Car/PDMS fiber (ranging from 3.2% to 14.7%). A linear regression model with a log-transformed response showed that masses extracted were not significantly different between the two SPME fibers (*p*-value = 0.44), as well as between both fibers and theoretical values (*p*-value = 0.43). The differences in mass extracted with 50/30-µm DVB/Car/PDMS at every sampling time were 9% less than the mass extracted with 85-µm Car/PDMS, respectively. Log-transformation of mass extracted on SPME fiber was performed because there was non-constant variance in the residuals.

### 2.3. Effect of the Glass Liner Conditioning

There was no apparent effect of the glass liner conditioning on sampling of acetic acid. TWA sampling of acetic acid using exposed SPME fibers inside cleaned (new, unused) and “saturated” (exposed to standard gas for an extended period) glass liners was carried out. The rationale for testing the “saturated” glass liner was to test if that kind of conditioning is needed for practical air sampling. Resulting total masses extracted on SPME fibers were reproducible. RSDs ranged from 4.3% to 8.2% with cleaned and from 1.6% to 7.9% with “saturated” liners. A two-sample *t*-test did not show a statistically significant difference in mass extracted on the SPME fiber exposed inside cleaned and “saturated” liners. To determine if the rates of increase were different, a linear regression model with a log-transformed response was used. Log-transformation of masses extracted on SPME fiber was performed because there was non-constant variance in residuals. Fitting of the model showed no significant difference in interaction between the condition of a glass liner and time (*p*-value = 0.74). Upon analyzing the means of mass extracted on an SPME fiber with different glass liners at each time point, the *p*-values were not significant (from 0.68 to 0.93 for each time point, respectively). However, one of the interesting findings was that the percentage difference between both glass conditions was the highest at a sampling time of 1 h (15.0%). Then, the percentage difference decreased to 2.6% at a sampling time of 4 h and continued to decrease at longer sampling times. Figure 4 summarizes the results of previous experiments with both SPME fibers, and both glass liner conditions (clean 85-µm Car/PDMS vs. “saturated”).

The result of the previous analysis shows that the SPME fibers extracted reproducible amounts of the target compound. Thus, the theoretical mass extracted on the fiber was proportional to the diffusion coefficient of the acetic acid, the concentration of the acetic acid in the gas phase, sampling time, and cross-sectional area of the glass liner opening, and it was inversely proportional to the diffusion path (i.e., distance between glass liner opening and the SPME fiber tip) length.

We also showed that the mass of extracted VOC on the SPME fiber remains inversely proportional to the retraction depth as a prerequisite for using Equation (1) for quantification. Thus, it was decided to investigate the possible influence of SPME fiber retraction depths inside a glass liner on extracted mass. Several diffusion path lengths (5, 10, 30, and 35 mm) were tested and compared to the fixed retraction depth of 17 mm that was used in the previous experiments. The aim of these new tests was to identify if different retraction depths would affect the mass extraction process inside of a glass liner. The results of the effect of retraction depth are shown in Figure 5.

Extracted masses at the diffusion path lengths followed a power-law distribution. RSDs for extracted masses did not exceed 10% (7.6, 2.0, 5.5, 1.7, and 3.3% for 5, 10, 17, 30, and 35 mm, respectively). Thus, a diffusion path length can be adjusted, e.g., for achieving lower detection limits and/or higher accuracies at higher sampling times [22].

### 2.4. Verification of Glass-Liner-Facilitated TWA SPME via a Side-by-Side Comparison with the Sorbent-Tube-Based Method

The new method was compared with sorbent-tube-based sampling (a conventional method). Table 1 shows the comparison of measured concentrations of acetic acid in indoor air (laboratory, office space) and at a commercial swine farm in Iowa. Triplicates were taken at each sampling site.

The concentration of acetic acid in the air was calculated using Fick’s first law of diffusion (Equation (2)):
(2)$$C_{gas}=\frac{m\cdot Z}{D_{g}\cdot t\cdot A}$$

Generally, the masses extracted by the SPME fibers were reproducible. In comparison with sorbent tubes, SPME fibers were much simpler to operate and did not require a thermal desorption system and additional instruments (a flowmeter and a pump) for VOC sampling in the field. It was also convenient to use in quiet places such as an office; the noise of the running pump caused a little discomfort to graduate students.

A comparison of the two methods showed that the concentrations obtained using the SPME fibers were much higher than the result based on the sorbent tubes. The difference between those methods varied depending on the sampling site. For example, in an indoor air setting, the differences between the two methods in resulting concentrations of acetic acid were 58% and 78% in the office and the laboratory, respectively. The differences between the two methods in the indoor setting were statistically significant (*p* < 0.002). Both indoor sampling sites had nearly similar concentrations of acetic acid. The small difference in concentrations between those two sampling sites could be explained by the more efficient ventilation system in the laboratory, which helped keep the concentration of the compound low, whereas the doors of the office were kept closed during the sampling, so there was less air mixing between the office and the hallway.

The TWA concentration of acetic acid in swine barns was approximately 50–200 times higher compared to indoor air environments. A sampling of acetic acid at Farm 1 for two days revealed larger differences in results produced by “retracted” SPME and sorbent tubes. The differences were statistically significant (*p* < 0.001). Sampling with tubes was much shorter over the entire period and, thus, not capable of measuring variations. During the first day of sampling, the glass tubes housing SPME fibers were placed in the direction facing the barn air flow. On that day, the differences between both methods were the highest (130%). On the following day, when SPME fibers were placed pointing in the direction of exhaust fans (i.e., glass liner opening faced the other direction), the discrepancies decreased (by nearly 26%), but remained high. The effect of TWA SPME sampler positioning requires additional research. An interesting fact is that the concentrations measured by the two methods were higher than previously reported in the literature. At Farm 2, both methods showed less differences than at the first farm. The differences between them did not exceed 70%. The RSDs of masses extracted for both methods were under 11%. In Table 1, at Farm 2, only one sample with sorbent tubes was taken on Day 1, so SD could not be calculated.

## 3. Materials and Methods

All materials and methods are described in greater detail by Tursumbayeva’s (2017) [34] graduate thesis. Below is a summary of key details.

### 3.1. Chemicals and Materials

Chemicals used in this study included acetic acid and helium. Acetic acid, glacial (certified by ACS (American Chemical Society) ≥ 99.7%) was purchased from Fisher Chemical (Fair Lawn, NJ, USA), and helium (≥99.99%) was purchased from Air Gas (Des Moines, IA, USA). The 85-µm Car/PDMS and 50/30-µm DVB/Car/PDMS SPME fibers and manual SPME holders were obtained from Supelco (Bellefonte, PA, USA).

### 3.2. Standard Gas Generation and Sampling System

The standard gas generation and sampling system were built to simulate typical air flow rates through swine facilities (Figure 6).

The standard gas generation system included sampling ports for air quality check, a mass flow controller (Aalborg, Orangeburg, NY, USA), a motorized syringe pump (KD Scientific, Holliston, MA, USA), a 50-µL gastight syringe (Hamilton, Reno, NV, USA), a mixing port, polytetrafluoroethylene (PTFE) tubing (Thermo Scientific, Rochester, NY, US), and compression fittings. After the clean compressed air was introduced into the standard gas generation system, it flowed through the air quality check to be purified. Air flow (150 mL·min^−1^) was managed by a mass flow controller. The rate of the target compound injection was controlled by a motorized syringe pump. Known volumes of the target compound were introduced to clean air in a heated mixing port to produce the desired concentrations. After standard gas (*C_gas_*) was generated, it passed through the gas sampling system.

The gas sampling system consisted of two U-shaped gas bulbs submerged inside of a thermostated water bath. Gas bulbs were filled with solid glass balls to help evenly distribute acetic acid in clean air. Both sides of the bulbs were sealed with lids. A sampling port was installed on one of the lids of a bulb. Sampling ports included an SPME fiber enclosed in a glass liner (Figure 6). The distance between the opening of the liner and the tip of the fiber was fixed at 1.75 cm. As can be seen in the inset of Figure 7A,B (close-up), a glass liner was inserted into the gas bulb. The PTFE tubing was slid around the top of the glass liner. A septum was inserted into the PTFE tubing to close the top of the glass liner and for SPME needle insertion. The water bath was covered with insulation material to avoid excessive water evaporation. The temperature of the water in the bath was held at 25 °C. After passing through the gas sampling system, air flow was checked with a volumetric flowmeter (Bios Defender 520, MesaLabs, Butler, NJ, USA) to detect possible leaks in the system, and then exhausted to the fume hood.

The mass flow controller and the motorized syringe pump were used to produce the desired concentration of acetic acid in the gas generation system. The maximum concentration of acetic acid (617 µg·m^−3^) which was reported by Cai et al. [11] was chosen in our research to assess the method. To achieve the desired concentration, the rate of acetic acid injection into a heated mixing port was calculated using Equations (4)–(6) described in the study by Baimatova et al. [30]. Since the calculated injection rate to generate 617 µg·m^−3^ acetic acid in the system was small (5.553 µg·h^−1^), it was decided to dilute acetic acid with distilled water at the ratio of 5 to 1000. The syringe with the acetic acid standard solution was refilled every day. The dilution with water also helped avoid big fluctuations in the concentration of acetic acid since the dilution increased the number of solution injections into the system (Figure 2). A description of quality assurance and quality control measured pertaining to the liquid injection and flow rate verification are provided in Appendix B.

### 3.3. MS Detector Calibration with an Acetic Acid Standard Solution

To convert the peak area count of acetic acid extracted from the SPME fiber, we needed to know the response factor. The response factor was obtained by injecting different volumes (0.1–0.3 µL) of the analyte solution in hexane into the GC inlet working in splitless mode and determining the corresponding peak areas. Direct injections were conducted in triplicate. The calibration curve was constructed using four data points of average masses of acetic acid (from 500 to 5000 ng) that were injected into the GC. Response factor was calculated from the average mass injections, and corresponding peak area counts (Equation (3)).
(3)$$RF=\frac{PA}{m}$$
where *RF* is the response factor, *PA* is the peak area count, and *m* is the known mass introduced into a column. Taking into account that the instrumental responses were linear over the tested period and the intercept was statistically zero, the response factor was equal to 14,400 peak area units·ng^−1^. Knowing the response factor, the quantification of acetic acid mass extracted on the SPME fiber was done using the same equation.

### 3.4. SPME Fiber Conditioning

A new SPME fiber was thermally cleaned in a heated GC injection port according to the manufacturer’s instructions. Before each sampling, the SPME fiber was cleaned in the GC injector port. This was done by holding the SPME fiber in the heated GC injection port at 240 °C for 3 min. Then, the fiber was introduced into the glass liner at the sampling port. After adsorption of the target compound, the SPME fiber was quickly transported to the GC injection port, where it was kept for 3 min for desorption. Between injections, the SPME fiber was kept in aluminum foil to avoid the absorption of VOCs present in the laboratory air.

### 3.5. Conditions of GC–MS

A gas chromatograph coupled with a mass spectrometer (6890N/5975C, Agilent, Santa Clara, CA, USA) was used in this study. Helium was selected as a carrier. The constant flow of helium in the column was 7.5 mL·min^−1^. The flow was relatively high for an MS because the instrument was fitted with an olfactometry port/open split interface (human panelists were not used in this research). Temperatures of the ion source, quadrupole, and MS interface were 230, 150, and 240 °C, respectively. Splitless mode on the GC injection port at 240 °C was used. The oven temperature was initially set at 40 °C for 3 min, followed by heating rate increments of 7 °C·min^−1^ up to 125 °C, and 30 °C·min^−1^ up to a final 240 °C (held for 2 min). Total GC run time was 29.41 min. The retention time of acetic acid was 12.7 min. The MS detector was autotuned daily.

### 3.6. Standard Gas Stability Check

The standard gas that was generated by the gas generation system was checked for stability. For this purpose, the standard gas was checked several times for three consecutive days. The standard gas was sampled with an SPME fiber every hour after injection with an exposed 85-µm Car/PDMS fiber. A sampling time of 20 s was sufficient. Simultaneously, the concentration of acetic acid was monitored with the same type of fiber, but in a “retracted” position. The sampling time for the “retracted” fiber was 1 h. This stability check provided the information that the system was capable of producing stable responses over time and that the data which were going to be collected in the future would be reproducible. Furthermore, before starting a new set of experiments, the concentration of acetic acid was verified with an exposed fiber. At the same time, the standard method (sorbent tubes) was used to verify the concentration of acetic acid in the system. After 24 h, the syringe was refilled with an acetic acid solution (50 µL).

### 3.7. Experimental Design

Calibration of the SPME fiber was conducted by exposing the fiber inside a glass liner to the air with an acetic acid concentration of 617 µg·m^−3^ at 25 °C generated by the standard gas generation system. Retraction depth was fixed at 1.7 cm. The inner diameter of the glass liner (a standard GC liner recommended for SPME injections) was measured using a digital microscope (CC-HDMI-CD1, New Haven, CT, USA) and was equal to 0.844 mm. As an adsorptive fiber [35], the SPME fiber required testing of different sampling times to make sure that the fiber did not reach its sorptive capacity. Thus, sampling times of 1, 4, 8, and 12 h were examined to determine the longest sampling time before the sorptive capacity limit of the fiber was reached. All experiments were completed in triplicate. To improve the S/N ratio, quantification of acetic acid was performed using SIM mode at *m*/*z* 60.0. Limits of detection (LOD) and quantification (LOQ) were calculated by estimating concentrations corresponding to signal-to-noise (S/N) ratios 3:1 and 5:1, respectively.

### 3.8. SPME Fiber Selection

Two commercially available SPME fibers, 85-µm Car/PDMS and 50/30-µm DVB/Car/PDMS, were tested to select the most suitable fiber for extracting the target compound. Both SPME fibers were inserted in each sampling port (Figure 6) and exposed inside a glass liner. Before every SPME fiber injection, glass liners were washed and baked overnight. Extractions of acetic acid with the two different fibers were conducted simultaneously. Three replicate samples were taken with each fiber. Sampling times between 1 and 12 h were examined. Constant dry air flow at 150 mL·min^−1^ with a diluted acetic acid injection rate of 5.55 µg·h^−1^ was used to generate the desired concentration.

### 3.9. Effect of Glass Liner

The possible effect of glass liner conditioning was examined because of the rationale based on previous studies of Baimatova et al. [30,31] and Koziel et al. [32], which accounted for adsorption to the SPME metallic assembly. In their work, SPME needle assembly was shown to extract a significant portion of VOCs. To minimize or possibly eliminate this effect, the exposed SPME fiber was inserted into a protective glass liner (Figure 1). Two different conditions of a glass liner were tested. In the “cleaned” condition, glass liners were washed and baked overnight to evaporate all remaining VOCs. Cleaned liners were inserted into the sampling port in the standard gas generation system immediately before the SPME fiber insertion. In the “saturated” condition, glass liners remained in the sampling port of the standard gas generation system for at least an hour before SPME fiber insertion. A *t*-distribution was used to test the null hypothesis that the two population means (mass extracted on the SPME fiber exposed to cleaned and saturated liners) had no statistical difference at the 95% confidence interval (CI) (two-tailed test).

### 3.10. Sorbent Tubes

Sorbent tubes packed with Tenax TA were used to compare the results of the exposed SPME fiber inside a glass liner. The sorbent-tube-based method was used as a “benchmark” for the new method. Table A1 summarizes the pros and cons of compared and available methods. The procedure of sampling with sorbent tubes was completed as described in the work of Zhang et al. [10]. Firstly, sorbent tubes were thermally cleaned at 260 °C under a 100-mL·min^−1^ N_2_ flow for 5 h; then, before subsequent uses, they were pre-conditioned at 260 °C under a 100-mL·min^−1^ N_2_ flow for 30 min. In the field, sorbent tubes with two sections, sampling and breakthrough (against saturation), were connected to an air sampling pump (SKC Inc., Eighty Four, PA, USA) at a 50 mL·min^−1^ set flow rate. The sampling flow rate was monitored with a flow meter.

### 3.11. Application in the Field

After validating the described method in the lab, sampling of acetic acid was performed in indoor and livestock settings. Indoor air sampling included two sites: a manure treatment laboratory and an office space at Iowa State University. In the livestock setting, air sampling of acetic acid was carried out inside of the barns. Livestock air samples were taken at two swine farms: a typical swine farm located in Central Iowa (Farm 1) and a new farm with air scrubber and filtration technology for odor reduction (Farm 2). Both the new method (i.e., an SPME fiber exposed inside of a glass liner) and the conventional method (i.e., the sorbent tubes) were used at the sampling sites. The samplers were placed upstream of exhaust fans. The opening of the “retracted” fibers and sorbent tubes were pointed in the direction of the exhaust fans.

Three 85-µm Car/PDMS fibers were used at each site. Every fiber was thermally cleaned in a GC injector port as described earlier. Then, the fiber was assessed for residuals. For SPME fiber protection in the field, a “retracted” SPME fiber was placed inside of a 40-mL thermally cleaned vial. This was done to make an additional barrier between the dusty and odorous environment and the TWA SPME sampler (Figure 8). Thus, only the opening of the glass liner was exposed to the environment. Vials with a “retracted” SPME fiber were kept in thermally clean aluminum foil to prevent any interaction with the environment before actual sampling. Depending on anticipated concentrations at each monitoring site, the sampling time for the “retracted” SPME fiber was adjusted. For the quantification of acetic acid in the indoor setting, a sampling time of 12 h was used. For testing the method in the livestock setting, a sampling time of 40 min was sufficient. The diffusion coefficient was equal to 1.1 × 10^−5^ m^2^·s^−1^ at 25 °C [36].

Quantification of acetic acid was also performed with Tenax sorbent tubes. The sorbent tubes were thermally cleaned as described earlier. Multiple air samples were taken with two adjacent sorbent tubes, and the results were averaged for the indoor setting. The sampling time was 20 min. For the swine farm setting, a sampling time of 40 min was used.

After samples were taken, SPME fibers and sorbent tubes were covered with thermally cleaned aluminum foil and placed in clean glass vials and then transported for further analysis. All samples were analyzed within 5 h of sample collection. A *t*-distribution was used to test the null hypothesis that the sample means received with the two methods were equal at the 95% CI (two-tailed test).

## 4. Conclusions

A novel and simple TWA SPME-based method for the quantification of acetic acid in ambient air was developed. The following conclusions can be drawn:
An SPME fiber exposed inside a glass liner followed Fick’s law of diffusion. There were linear relationships between mass of the analyte extracted and extraction time up to 12 h (*R*^2^ > 0.99), and mass extracted and the inverse of retraction depth (1/*Z*) (*R*^2^ > 0.99). The amount of VOC adsorbed via the TWA SPME using a GC glass liner to protect the SPME was reproducible.There was no statistically significant difference between cleaned and “saturated” (equilibrated) glass liners. Thus, no special precautions are recommended for a practical application of this approach.The 85-µm Car/PDMS fiber revealed a higher response than the DVB/Car/PDMS fiber. The mass extracted by Car/PDMS was 8.9% higher than the mass extracted by the DVB/Car/PDMS fiber coating.The limit of detection (LOD, S/N = 3) and limit of quantification (LOQ, S/N = 5) were 10 and 18 µg·m^−3^ (4.3 and 7.2 ppbV), respectively.The new method was evaluated under field conditions by comparing it to the standard method (sorbent tubes) in four different locations. The TWA SPME sampling with a glass liner showed a reasonable match with the sorbent tubes.

The method shown is a relatively simple and practical, yet accurate sampling technique for the quantification of acetic acid in both an indoor workplace and a swine farm building. The method is reusable. Further research should be done to extend the number of odorous VOCs that can be used with this method, allowing further improvement of TWA SPME modeling (e.g., Reference [22]), and the incorporation of temporal changes in sampled air on TWA SPME [37].

## Figures and Tables

**Figure 1 molecules-24-00406-f001:**
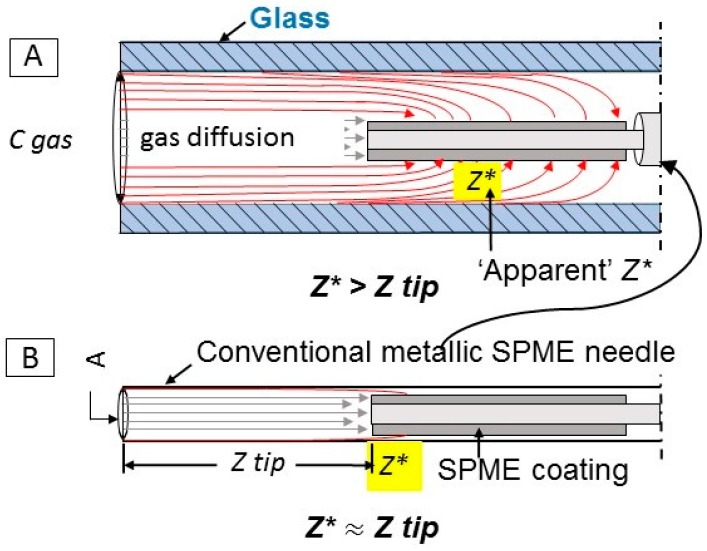
Time-weighted average (TWA) gas sampling with solid-phase microextraction (SPME). Comparison of proposed (**A**) and conventional (**B**) TWA SPME. (**A**) Sampling with SPME fiber exposed and retracted inside of a glass liner; (**B**) a typical case of TWA SPME where the SPME fiber is retracted inside of a conventional SPME needle. Gray arrows represent the diffusion path between bulk gas (left side) and the retracted fiber tip (Z tip). Red arrows represent the “apparent” diffusion path extending beyond the tip to the SPME fiber coating side. The “apparent” diffusion path represents the extracting process enhanced by the sides of the SPME coating. *Z** may continue to increase after the tip is saturated.

**Figure 2 molecules-24-00406-f002:**
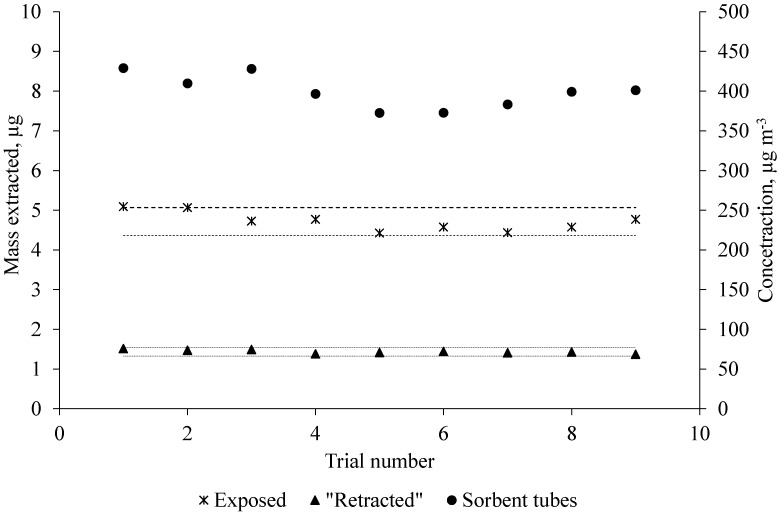
Standard gas stability needed for simulating steady-state conditions for TWA SPME sampling. Extraction conditions: two 85-µm Car/PDMS SPME fibers (one was a standard exposed fiber, and the other was an exposed fiber that was kept inside of a glass liner). Both were exposed to the standard gas (acetic acid, *C_gas_* = 617 µg·m^−3^). Retraction depth was 17.5 mm. Gas sampling was performed every hour for three consecutive days. Sampling times were 20 s for the exposed SPME fiber and 1 h for the retracted SPME fiber. The dashed lines on the graph indicate a ±7.5% band from the average. The concentration of acetic acid was verified with sorbent tubes. The concentration of acetic acid in the system obtained by sorbent tubes is shown on the right *y*-axis. Selected ion monitoring (SIM) mode at *m*/*z* 60.0 was used for acetic acid detection and quantification.

**Figure 3 molecules-24-00406-f003:**
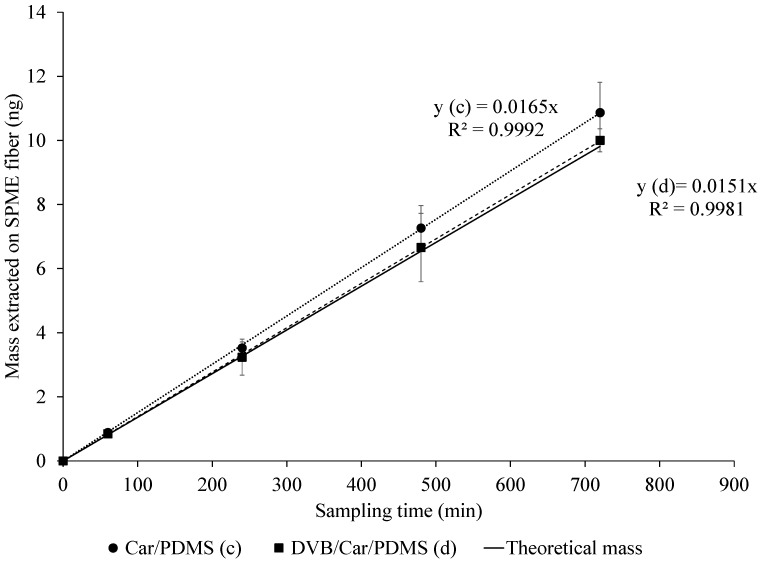
TWA SPME where fiber is retracted into a glass liner. Comparison of the extraction efficiency of acetic acid by 85-µm Car/PDMS and 50/30-µm DVB/Car/PDMS SPME fibers. The theoretical mass on the SPME fiber (shown as a solid line) was calculated using Equation (1) (Fick’s law of diffusion). The experimental masses are shown as dotted and dash lines for Car/PDMS and DVB/Car/PDMS fibers, respectively. Extraction conditions: 85-µm Car/PDMS fiber exposed inside a glass liner, standard gas (acetic acid, *C_gas_* = 617 µg·m^−3^). Retraction depth was 1.75 cm. SIM mode at *m*/*z* 60.0 was used for acetic acid detection and quantification. Experiments were completed in triplicate.

**Figure 4 molecules-24-00406-f004:**
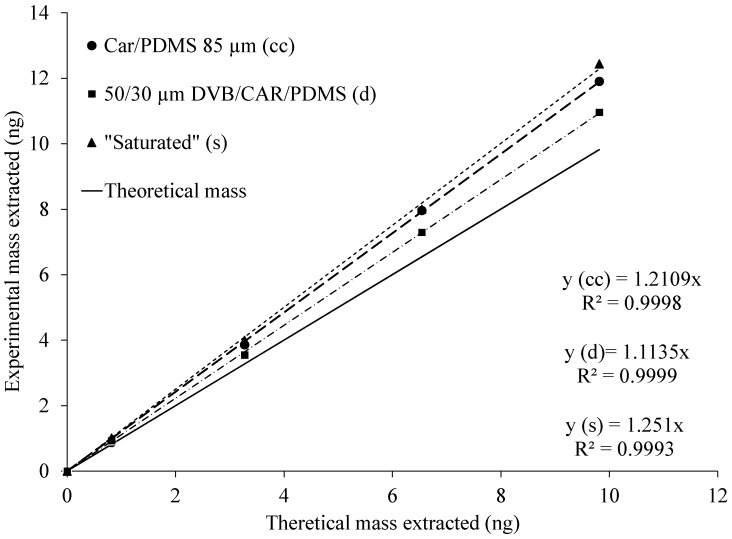
Comparison of the theoretical mass with the experimental masses extracted using 85-µm Car/PDMS (with clean and saturated glass liners) and 50/30-µm DVB/Car/PDMS fibers. The theoretical mass extracted was calculated using Fick’s first law of diffusion (Equation (1)).

**Figure 5 molecules-24-00406-f005:**
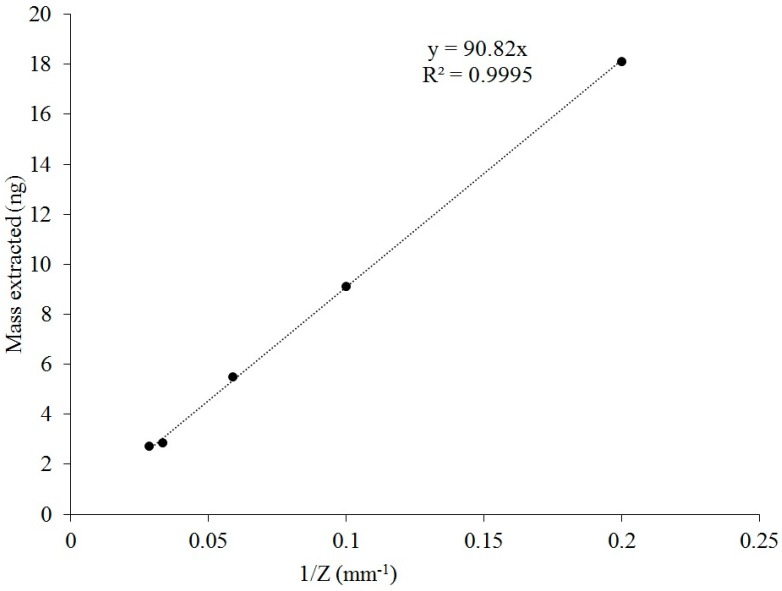
Effect of diffusion path length (*Z*) on the extracted mass of acetic acid. Extraction conditions: “retracted” 85-µm Car/PDMS, standard gas (acetic acid). SIM mode at *m*/*z* 60.0 was used for detection and quantification of the target compound. A sampling time of 4 h was used.

**Figure 6 molecules-24-00406-f006:**
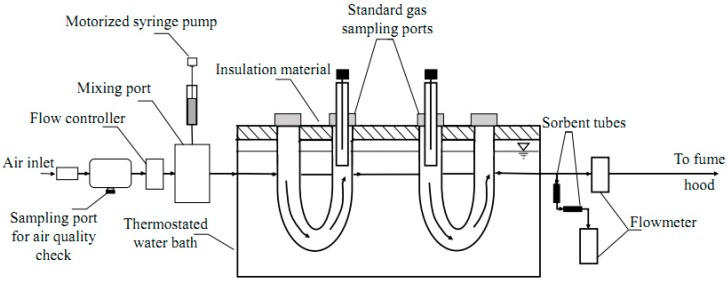
Schematic of standard mixture flow in the system. Passive gas sampling was completed with SPME retracted inside a gas chromatography (GC) injector glass liner.

**Figure 7 molecules-24-00406-f007:**
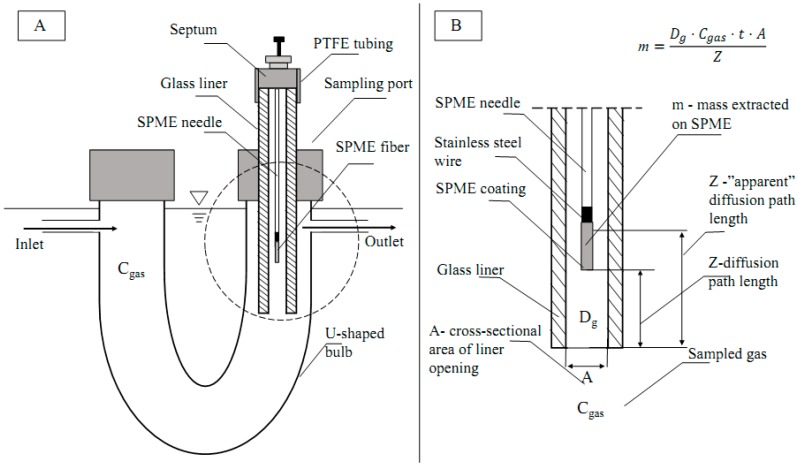
Passive gas sampling with SPME fiber retracted inside a GC injector glass liner. (**A**) Design of sampling port in the standard gas generation system. (**B**) Terms in Fick’s first law of diffusion used for quantifications. The SPME fiber is exposed inside of a GC glass liner; thus, the walls of the liner serve as a protective needle in the conventional retracted mode.

**Figure 8 molecules-24-00406-f008:**
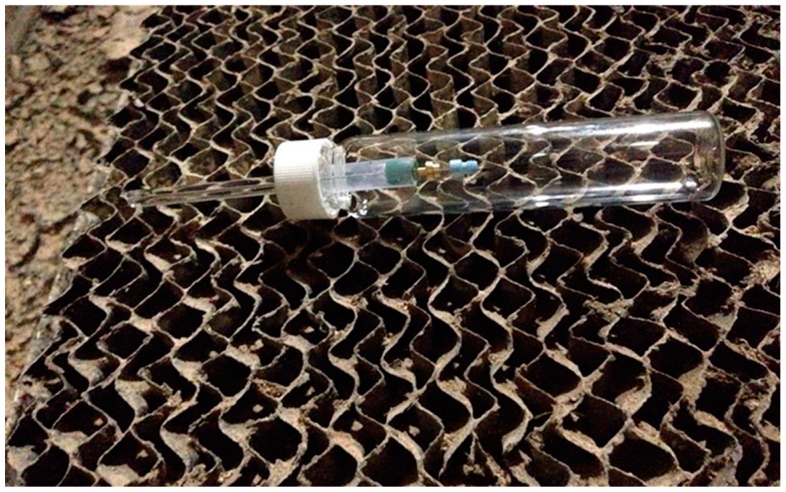
Field air sampling in TWA SPME mode on a commercial swine farm (Farm 2) in Iowa. The air sample diffuses through the opening in the GC glass liner (left side of the photo) onto an SPME fiber fully exposed inside the liner. A short section of Teflon tubing and a half-hole septum seal the liner and facilitate SPME insertion. A clear glass vial encloses the SPME assembly from dust and other gases in the sampled air.

**Table 1 molecules-24-00406-t001:** Comparison of measured acetic acid concentrations in different locations using time-weighted average (TWA) solid-phase microextraction (SPME) (85 µm Car/PDMS) facilitated with a glass liner and sorbent-tube-based measurement.

Location	Measured Concentration (µg·m^−3^)		% Difference (TWA SPME vs. Sorbent Tubes)	*p*-Value
	TWA SPME (Glass Liner)	Sorbent Tubes		
Office	17.7 (±2.7)	9.7 (±1.0)	58	0.002
Laboratory	15.2 (±0.8)	6.6 (±0.7)	78	0.0001
Farm 1, Day 1	3620 (±430)	755 (±20)	131	0.0004
Farm 1, Day 2	2400 (±310)	750 (±180)	104	0.0008
Farm 2, Day 1	685 (±70)	340	67	0.002
Farm 2, Day 2	750 (±90)	375 (±20)	67	0.0001

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
