# Peer review of "Development of Time-Weighted Average Sampling of Odorous Volatile Organic Compounds in Air with Solid-Phase Microextraction Fiber Housed inside a GC Glass Liner: Proof of Concept"

_molecules, 2019, doi:10.3390/molecules24030406_

Round 1

Reviewer 1 Report

The manuscript describes the proof of concept of a method for the determination of VOCs using TWA-SPME. Although the title suggests a work carried out taking into account a set of analytes, in reality, the work is based only on the determination of acetic acid chosen as a model compound.

The work appears as a continuation of an article previously published on "Molecules". Overall the study was well presented and the manuscript is written with fluent English. In my opinion, there are no significant flaws concerning the research, the only problem concerns the editorial placement. In fact, while recognizing the original points of the work carried out, the idea of using a GC liner for TWA-sampling has already been proposed in the literature by other authors and also discussed in the previous article (Kenessov et al. Molecules 2018, 23 (11), 2736 ). Moreover, since the study was carried out with only one analyte (acetic acid), its feasibility in the analysis of VOCs is still very far from being demonstrated.

Minor remarks:

Paragraph 1.3 can be summarized by avoiding much of the literature mentioned.

Paragraph 3.3 could be deleted.

In MDL and LOQ calculation, are the samples used to calculate the standard deviation blank samples? Please clarify this point.

What was the value of the acetic acid diffusion coefficient? How was it determined? I suggest specifying this information in the text. Furthermore, although the Fick’s equation is reported, it is useful to insert also the final analytical equation used to determine the acetic acid concentrations of Table1.

Author Response

Comments and Suggestions for Authors

The manuscript describes the proof of concept of a method for the determination of VOCs using TWA-SPME. Although the title suggests a work carried out taking into account a set of analytes, in reality, the work is based only on the determination of acetic acid chosen as a model compound.

The work appears as a continuation of an article previously published on "Molecules". Overall the study was well presented and the manuscript is written with fluent English. In my opinion, there are no significant flaws concerning the research, the only problem concerns the editorial placement. In fact, while recognizing the original points of the work carried out, the idea of using a GC liner for TWA-sampling has already been proposed in the literature by other authors and also discussed in the previous article (Kenessov et al. Molecules 2018, 23 (11), 2736 ). Moreover, since the study was carried out with only one analyte (acetic acid), its feasibility in the analysis of VOCs is still very far from being demonstrated.

Author’s Response:

We appreciate Reviewer comments.  The Reviewer comment about ‘other authors’ is well taken – we are the ‘other authors’.  This has a been a collaborative work for nearly a decade with several papers focused on advancing the TWA-SPME concept for air sampling and analysis.  This manuscript is based on Ms. Madina Tursumbayeva M.S. thesis (cited as reference [34] doi: 10.31274/etd-180810-5245 which was submitted/published in May 2017, i.e., much earlier than Kenessov et al. Molecules 2018, 23 (11), 2736, reference [22].  We were not able to turn the M.S. thesis into a manuscript earlier. 

It is a ‘proof-of-concept’ manuscript as indicated in the title for a large and important class of air pollutants (odorous VOCs) that are overlooked by researchers working with mainly urban air pollution and for which there are no reliable standard methods. 

We described the rationale for choosing acetic acid in the last paragraph before section 1.1. as being considered the most abundant VOC in any animal facility, including swine farms. We also stated (e.g., Abstract) that acetic acid was chosen as a model compounds for odorous VOCs.   

Minor remarks:

Paragraph 1.3 can be summarized by avoiding much of the literature mentioned.

Author’s Response:

Paragraph 1.3 was summarized and many details were deleted.

Paragraph 3.3 could be deleted.

Author’s Response:

The entire section 3.3 was moved to a new Appendix B. 

In MDL and LOQ calculation, are the samples used to calculate the standard deviation blank samples? Please clarify this point.

Author’s Response:

In our research, MDL and LOQ were calculated using equations (3) and (4) in the manuscript at the concentration of 617 µg m-3.  In addition to that first method, LOD and LOQ were estimated as concentrations corresponding to signal-to-noise (S/N) ratios 3:1 and 5:1, respectively.  This information for both methods was added to the manuscript.       

What was the value of the acetic acid diffusion coefficient? How was it determined? I suggest specifying this information in the text. Furthermore, although the Fick’s equation is reported, it is useful to insert also the final analytical equation used to determine the acetic acid concentrations of Table1.

Author’s Response:

The diffusion coefficient that was used in this study was equal to 1.1×10-5 m2/s at 25 °C. This coefficient was taken from the following manuscript that was not mentioned in the references. The article name:

Hafner, S.D.; Montes, F.; Rotz, C.A. Modeling Emissions of Volatile Organic Compounds from Silage. ASABE 2009, 3, 1895-1911.

This new reference was added to the manuscript.

Fick’s equation was added to the manuscript (as equation 2).

Reviewer 2 Report

The present manuscript deals with the development of a new device for the improved sampling (TWA) of odorous compounds in air, more specifically in atmosphere of swine facilities. In my opinion, although  this study has been limited to acetic acid, the research has been properly designed and the results are of interest, especially taking into account the lack of simple and reliable methods for on-site tests in this kind of samples.  However, I suggest the authors to introduce some minor changes before publication:

-          In my opinion, the Introduction contains quite basic information, particularly the description of the sampling methods with bags, canisters, sorbent tubes and with SPME fibers. I think that this information could be moved to Table A1.

-          Caption to Figure 4. From my understanding, a caption should give the information necessary for the readers to understand what is depicted in the figure, while the interpretation and discussion of the results should be placed in the text.  I suggest the authors to change the caption to this figure, removing the two first sentences.

-          Lines 418-425. The authors calculated the response factor for each individual mass of acetic acid injected, for different  masses. Usually, the instrumental responses are fitted to calibration lines. Did the authors check if the responses were linear over the tested mass interval? If so, was the intercept statistically zero? In my experience, calibration lines in chromatography often show non-zero intercepts. If this was the case, the equation of the calibration line would be a better option for calculating the acetic acid mass extracted than the expression (2).  This should be clarified in the text.

-           Lines 430-431. It is stated that after sampling the SPME fiber was quickly transported to the GC equipment, even though apparently this section refers to assays carried out in the lab. However, when the method was applied in the field (section 3.12) nothing is said about conditions used for transportation and storage of the fibers until analysis. This information has to be added in the text.    

Author Response

Comments and Suggestions for Authors

The present manuscript deals with the development of a new device for the improved sampling (TWA) of odorous compounds in air, more specifically in atmosphere of swine facilities. In my opinion, although this study has been limited to acetic acid, the research has been properly designed and the results are of interest, especially taking into account the lack of simple and reliable methods for on-site tests in this kind of samples.  However, I suggest the authors to introduce some minor changes before publication:

 -          In my opinion, the Introduction contains quite basic information, particularly the description of the sampling methods with bags, canisters, sorbent tubes and with SPME fibers. I think that this information could be moved to Table A1.

 Author’s Response:

The details were moved as a commentary text ahead of Table A1 inside the Appendix A.

-          Caption to Figure 4. From my understanding, a caption should give the information necessary for the readers to understand what is depicted in the figure, while the interpretation and discussion of the results should be placed in the text.  I suggest the authors to change the caption to this figure, removing the two first sentences.

 Author’s Response:

The two first sentences were removed from the caption

-          Lines 418-425. The authors calculated the response factor for each individual mass of acetic acid injected, for different  masses. Usually, the instrumental responses are fitted to calibration lines. Did the authors check if the responses were linear over the tested mass interval? If so, was the intercept statistically zero? In my experience, calibration lines in chromatography often show non-zero intercepts. If this was the case, the equation of the calibration line would be a better option for calculating the acetic acid mass extracted than the expression (2).  This should be clarified in the text.

 Author’s Response:

The responses were linear over the tested mass interval and the intercept was statistically zero. This clarification was added to the text.

-           Lines 430-431. It is stated that after sampling the SPME fiber was quickly transported to the GC equipment, even though apparently this section refers to assays carried out in the lab. However, when the method was applied in the field (section 3.12) nothing is said about conditions used for transportation and storage of the fibers until analysis. This information has to be added in the text.    

Author’s Response:

The information is added in section 3.12.

Round 2

Reviewer 1 Report

Dear authors, thank you for improving the manuscript following my suggestions. Again, I think that the study is well presented and the manuscript is written with fluent English. In my opinion, there are no significant flaws concerning the research.

Comments:

Line 350: there are 2 “were”.

Are you sure that the followed guidance for MDL assesment is by EPA (ref.36)? 

The MDL (22 µg m-3) need to be recomputed. Indeed, according to the used procedure, the spike at 617 µg m-3 does not comply with the inequalities:  Calculated MDL < Spike Level < 10 x Calculated MDL. According to the cited guidance, if these conditions are not met, it is necessary to recalculate the MDL.

Author Response

Comments and Suggestions for Authors

Dear authors, thank you for improving the manuscript following my suggestions. Again, I think that the study is well presented and the manuscript is written with fluent English. In my opinion, there are no significant flaws concerning the research.

Comments:

Line 350: there are 2 “were”.

Authors Response: we deleted the 2nd ‘were’. Thank you.

Are you sure that the followed guidance for MDL assesment is by EPA (ref.36)? The MDL (22 µg m-3) need to be recomputed. Indeed, according to the used procedure, the spike at 617 µg m-3 does not comply with the inequalities:  Calculated MDL < Spike Level < 10 x Calculated MDL. According to the cited guidance, if these conditions are not met, it is necessary to recalculate the MDL.

Authors Response: Thank you for pointing this out.  Indeed, the MDL calculation does not comply with the ‘common sense’ check of conditions in reference 36. Thus, we deleted the erroneously calculated MDLs, reference [36], and the values of MDLs calculated with [36] in the Abstract, Results, Methods and Conclusions.  We made minor changes in text to report the LOD and LOQ using the S/N approach.  We made corrections to reflect that in the Abstract, Results, Methods, and Conclusions.  Reference numbers for the last two items were changed accordingly. Again, we are thankful for pointing this out.